# Action does not enhance but attenuates predicted touch

Xavier Job[1], Konstantina Kilteni[1,2]*

[1]Department of Neuroscience, Karolinska Institute, Stockholm, Sweden; [2]Donders Institute for Brain, Cognition and Behaviour, Radboud University, Nijmegen, Netherlands

**Abstract** Dominant motor control theories propose that the brain predicts and attenuates the somatosensory consequences of actions, referred to as somatosensory attenuation. Support comes from psychophysical and neuroimaging studies showing that touch applied on a passive hand elicits attenuated perceptual and neural responses if it is actively generated by one's other hand, compared to an identical touch from an external origin. However, recent experimental findings have challenged this view by providing psychophysical evidence that the perceived intensity of touch on the passive hand is enhanced if the active hand does not receive touch simultaneously with the passive hand (somatosensory enhancement) and by further attributing attenuation to the double tactile stimulation of the hands upon contact. Here, we directly contrasted the hypotheses of the attenuation and enhancement models regarding how action influences somatosensory perception by manipulating whether the active hand contacts the passive hand. We further assessed somatosensory perception in the absence of any predictive cues in a condition that turned out to be essential for interpreting the experimental findings. In three pre-registered experiments, we demonstrate that action does not enhance the predicted touch (Experiment 1), that the previously reported 'enhancement' effects are driven by the reference condition used (Experiment 2), and that self-generated touch is robustly attenuated regardless of whether the two hands make contact (Experiment 3). Our results provide conclusive evidence that action does not enhance but attenuates predicted touch and prompt a reappraisal of recent experimental findings upon which theoretical frameworks proposing a perceptual enhancement by action prediction are based.

*For correspondence: konstantina.kilteni@ki.se

**Competing interest:** The authors declare that no competing interests exist.

## Editor's evaluation

While decades of research findings have supported the idea that action attenuates predicted touch, a recent finding has countered this, proposing that action actually enhances predicted touch and that the previously observed attenuation is due to tactile contact. The present study rigorously probes these alternative hypotheses with three pre-registered experiments. They show that previous enhancement findings are due to the lack of a proper baseline condition of externally generated sensation with which to compare. In light of the recent opposing findings, the current paper is important and presents compelling evidence in favor of the sensory attenuation theory.

## Introduction

Dominant motor control theories propose that the brain uses an internal forward model in combination with a copy of the motor command (efference copy) to predict the sensory consequences of our movements (*McNamee and Wolpert, 2019*; *Shadmehr et al., 2008*; *Wolpert and Flanagan, 2001*). For example, the brain predicts the upcoming touch as one reaches toward an object. These predictions allow for the correction of motor errors without relying on sensory feedback that suffers from

intrinsic delays (*Shadmehr et al., 2010*), thereby improving the estimation of the current state of our body by combining the predicted touch with the actual sensory input (*Scott, 2004*; *Shadmehr et al., 2008*; *Todorov and Jordan, 2002*). These predictions *attenuate* the perception of the self-generated input (sensory reafference) compared to that of externally generated input (*Davidson and Wolpert, 2005*; *Franklin and Wolpert, 2011*; *Wolpert and Kawato, 1998*) and allow to infer whether the cause of the sensory input is the self or the environment (*Brown et al., 2013*; *Corlett, 2020*; *Idei et al., 2022*). A classic example of this attenuation is that we are unable to tickle ourselves with our own touch, yet we are easily tickled by the touch of others (*Blakemore et al., 2000a*). The attenuation of sensory reafference – also referred to as sensory cancelation – is thought to be necessary to compensate for the limited capacity of the sensory systems by optimally prioritizing the perception of more informative externally generated stimuli (*Bays and Wolpert, 2008*; *McNamee and Wolpert, 2019*). Thus, the attenuation model proposes that we dampen perceptual representations of expected self-generated stimuli to reduce redundancy and to highlight behaviorally relevant unexpected externally generated stimuli.

In contrast, an alternative theoretical framework proposes that predictions, including those arising from our motor commands, should not attenuate but instead enhance sensory signals, thereby allowing for sharper (i.e. more accurate) representations of predicted compared to unpredicted sensory events (*Press et al., 2020*; *Press and Yon, 2019*; *Yon et al., 2021*). This *enhancement* account – also referred to as the sharpening account – posits that predictions based on our motor commands are equivalent to expectations formed by statistical regularities in sensory input or from prior knowledge (e.g. at the North Pole, one expects to see a polar bear rather than an elephant) and that these predictions should bias our perception towards our expectations. The proposal mainly stems from experimental research outside the domain of action and argues that weak, noisy, or ambiguous sensory input that is in line with prior expectations should be enhanced to achieve, on average, more accurate representations. For example, we are more biased to report the presence of visual events that are statistically likely to occur rather than unlikely events (*Chalk et al., 2010*; *Wyart et al., 2012*), more sensitive to low-level visual features that are in line with prior expectations (*Stein and Peelen, 2015*; *Teufel et al., 2018*), and show greater biases when perceiving visual events that are congruent with our expectations (*Hudson et al., 2016a*; *Hudson et al., 2016b*). Such effects are thought to result from mechanisms that increase the 'gain' of expected information by altering the weights of different sensory signals (*Summerfield and de Lange, 2014*). Thus, the enhancement model proposes that we amplify the perceptual representations of expected compared to unexpected sensory input.

In the somatosensory domain, evidence supporting the attenuation model has repeatedly shown that touch delivered to one (passive) hand by the other (active) hand (i.e. self-generated touch) is perceived as weaker or less ticklish (*Asimakidou et al., 2022*; *Bays et al., 2006*; *Bays et al., 2005*; *Blakemore et al., 1999*; *Kilteni et al., 2021*; *Kilteni et al., 2020*; *Kilteni et al., 2019*; *Kilteni et al., 2018*; *Kilteni and Ehrsson, 2022*; *Kilteni and Ehrsson, 2017a*; *Kilteni and Ehrsson, 2017b*; *Knoetsch and Zimmermann, 2021*; *McNaughton et al., 2023*; *Weiskrantz et al., 1971*; *Wolpe et al., 2016*) and evokes attenuated somatosensory cortical activation (*Blakemore et al., 1998a*; *Hesse et al., 2010*; *Kilteni and Ehrsson, 2020*; *Shergill et al., 2014*; *Shergill et al., 2013*) compared to a touch of identical intensity applied on the passive hand that is externally generated. In contrast, evidence supporting the enhancement model has shown that the action of the active hand results in an increase in the perceived intensity of touch on the passive hand, provided that the active hand never receives touch simultaneously with the passive hand (i.e. hands do not make contact) (*Thomas et al., 2022*). This enhancement finding has been recently used to support the sharpening model and to argue that attenuation effects are due to unspecific gating processes caused by the simultaneous double tactile stimulation of the two hands rather than action prediction (*Press et al., 2023*).

The attenuation (or cancellation) and enhancement (or sharpening) models present strikingly different hypotheses regarding how action influences the perception of sensory input and are supported by contradictory experimental evidence, leading to debates between researchers (*Corlett, 2020*; *Fuehrer et al., 2022*; *Kilteni and Ehrsson, 2022*; *Press et al., 2023*; *Press et al., 2020*; *Thomas et al., 2022*). Clarifying how predictions about the sensory consequences of our movements affect our perception is fundamental to understanding the interaction between perception and action (*McNamee and Wolpert, 2019*; *Shadmehr et al., 2008*; *Wolpert and Flanagan, 2001*) but also for clinical and neurobiological theories of psychosis spectrum disorders, such as schizophrenia

(*Blakemore et al., 2000b*; *Corlett et al., 2019*; *Frith, 2005a*; *Frith, 2005b*; *Frith, 2019*; *Frith et al., 2000*; *Shergill et al., 2014*; *Shergill et al., 2005*) and schizotypy (*Asimakidou et al., 2022*), as well as functional movement disorders (*Pareés et al., 2014*), Parkinson's disease (*Wolpe et al., 2018*), and aging (*Wolpe et al., 2016*).

The present study aimed to contribute to this debate by directly contrasting the enhancement and attenuation models within the same paradigm. To this end, the same force discrimination task employed in earlier studies reporting attenuation (*Asimakidou et al., 2022*; *Bays et al., 2006*; *Bays et al., 2005*; *Kilteni et al., 2023*; *Kilteni et al., 2021*; *Kilteni et al., 2020*; *Kilteni et al., 2019*; *Kilteni and Ehrsson, 2022*; *Timar et al., 2023*) and enhancement (*Thomas et al., 2022*) was used to determine (a) whether movement of the right hand enhances the perceived magnitude of touch applied on the left hand when the two hands do not make contact (Experiments 1 and 2) and (b) whether attenuation effects are due to double tactile stimulation caused by the contact of the two hands or action prediction (Experiment 3). Capitalizing on the fact that *any* conclusion about whether action prediction 'attenuates' or 'enhances' the perception of the somatosensory input needs to be made with a comparison to one's somatosensory perception in the absence of action, we also included a condition in which participants passively received externally generated touch, with which we compared the participants' perception in all experimental conditions. Consequently, if participants perceive a touch as less or more intense during action than in the absence of action, we can infer that the received touch was attenuated or enhanced, respectively. This is a critical methodological novelty compared to previous studies (*Bays et al., 2006*; *Thomas et al., 2022*) because if such baseline conditions are missing, the same patterns of results can be incorrectly attributed to 'attenuation' or 'enhancement': for example, if one condition produces less attenuation than another, it may be interpreted as enhancement, and vice versa. Indeed, it was this baseline condition that turned out to be critical for distinguishing between attenuation and enhancement effects. All studies and analysis plans were pre-registered on the Open Science Framework prior to data collection (**Methods**).

## Results

### Experiment 1. Action does not enhance predicted touch

Thirty naïve participants moved their right index finger towards their left index finger to generate the touch on their left index finger with (*contact* condition) or without (*no-contact* condition) simultaneous stimulation on their active finger. A *baseline* condition in which the participants did not move their right index finger and received touch on the left index finger passively (externally generated touch) was included in order to distinguish between effects of attenuation or enhancement (*Figure 1A*).

During the force-discrimination task, participants judged the intensity of a '*test*' force and a '*comparison*' force (100 ms duration each) separated by a random interval ranging from 800 ms to 1200 ms. The forces were delivered to the pulp of the left index finger by a cylindrical probe attached to a lever driven by a DC electric motor. In each trial, participants reported which force they felt was stronger (the *test* or the *comparison*). The intensity of the *test* force was fixed at 2 N, while the intensity of the *comparison* force was systematically varied among seven force levels (1, 1.5, 1.75, 2, 2.25, 2.5, or 3 N). In the *contact* condition, participants moved their right index finger towards their left index finger after an auditory cue and actively tapped on a force sensor placed on top of, but not in direct contact with, the probe. The participant's active tap on the force sensor triggered the *test* force on their left index finger, thereby producing simultaneous stimulation of both fingers in the *contact* condition and the sensation of pressing with the right index finger against the left index finger through a rigid object. Participants were told that they would always make contact between their fingers (through the probe) in this condition. In the *no-contact* condition, following the same auditory cue, participants moved their right index finger towards their left index finger. At the beginning of the condition, the force sensor was removed and replaced with a distance sensor that detected the relative distance of their active finger as it approached their left index finger to trigger the *test* force, similar to *Bays et al., 2006* and *Thomas et al., 2022*. The distance threshold was set such that the position of the right index finger when triggering the *test* force was equivalent to that in the *contact* condition. Participants were told that they would never make contact between their fingers. Before the experiment, participants were trained to make similar movements with their right index finger in the *contact* and *no-contact* conditions. Finally, in the *baseline* condition (externally generated touch),

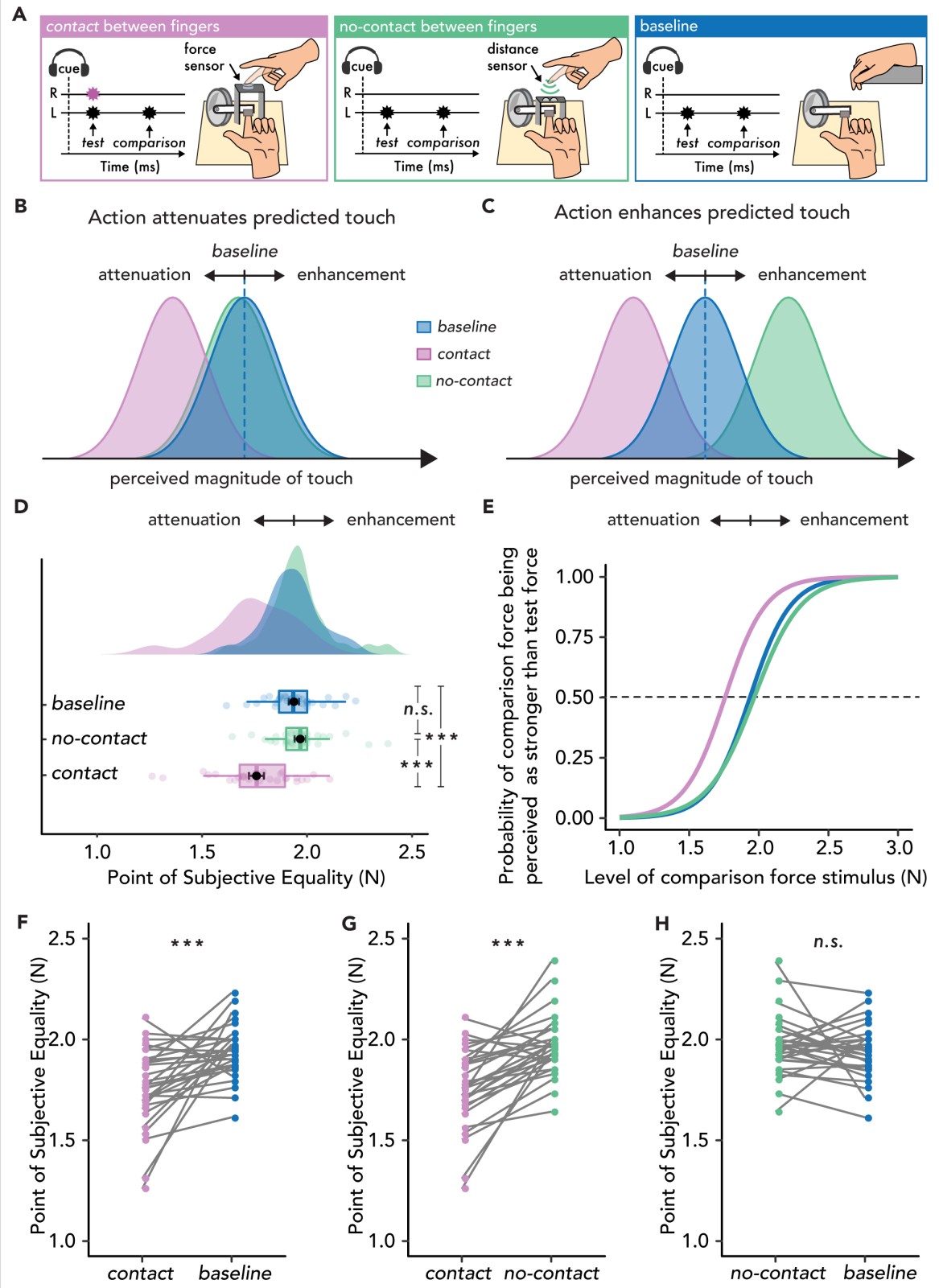

**Figure 1.** Experimental methods, hypotheses, and results of Experiment 1 (n=30). (**A**) In the *contact* condition (magenta), participants tapped with their right index finger 'R' on a force sensor placed above the probe that delivered a *test* force of 2N to their left index finger 'L', followed by a *comparison* force randomly varying between 1 and 3N. In the *no-contact* condition (green), participants approached a distance sensor with their right index finger, which triggered the test force on their left index finger, thus receiving no touch on their active finger. In the *baseline* condition (blue), participants

*Figure 1 continued on next page*

*Figure 1 continued*

relaxed both hands and passively received the forces on their left index finger. (**B**) Hypotheses based on the attenuation model. Touch in the *contact* condition (magenta) should be perceived as weaker than in the *baseline* (blue), but touch in the *no-contact* condition (green) should be perceived similarly to that in the *baseline* (blue). (**C**) Hypotheses based on the enhancement model. Touch in the *no-contact* condition (green) should be perceived as stronger than in the *baseline* (i.e. enhanced) (blue), but touch in the *contact* condition (magenta) should be perceived as weaker than the *baseline* (blue). Note that attenuation effects in the *contact* condition are predicted both by the attenuation and the enhancement model with the difference that the attenuation model attributes these effects to action prediction, while the enhancement model attributes these effects to the simultaneous touch on the active hand. (**D–H**) Data are color-coded per condition. (**D**) Box plots show the median and interquartile ranges for the point of subjective equality (PSE) values per condition, black circles and error bars show the mean PSE ± SEM, and the raincloud plots show the individual PSE values and their distributions. No enhancement effects were observed in the *no-contact* condition. (**E**) Group psychometric functions for each condition. The leftward shift of the curve in the *contact* condition indicates attenuated PSE values compared to the other two conditions. (**F–H**). Line plots for individual participant PSE values illustrate significantly lower PSEs for the *contact* versus *baseline* (**F**) and *no-contact* (**G**) conditions, but no significant differences in the PSE values between *no-contact* and *baseline* (**H**). (***$p<.001$, n.s. non-significant).

The online version of this article includes the following figure supplement(s) for figure 1:

**Figure supplement 1.** Fitted logistic models based on the participants' responses under each condition of Experiment 1.

participants were told to relax their right hand, and each trial began with the same auditory cue followed by the *test* force delivered to the participants' left index finger 800 ms after the cue. In all trials, the *comparison* force was delivered after the *test* force, independently of the right hand, and participants verbally reported their judgment (**Methods**).

Participants' responses were fitted with logistic psychophysics curves, and we extracted the point of subjective equality (PSE), which represents the intensity of the comparison force at which the test force feels equally as strong (***Figure 1—figure supplement 1***). Consequently, a PSE in a movement condition (*contact* or *no-contact*) that is lower than the PSE of the *baseline* condition indicates attenuation, while a PSE that is higher than the PSE of the *baseline* condition indicates enhancement (**Methods**). According to the attenuation model (***Figure 1B***), attenuation of the somatosensory input on the left hand in the *contact* condition compared to the *baseline* condition should be observed (i.e. lower PSEs) due to the *test* force being a predicted consequence of the action of the right hand. However, regarding the *no-contact* condition, earlier studies have shown that the mere movement of one hand is not sufficient to produce predictions of somatosensory input simultaneously applied on the other hand (***Bays et al., 2005***; ***Kilteni et al., 2018***; ***Kilteni and Ehrsson, 2020***; ***Kilteni and Ehrsson, 2017a***; ***Kilteni and Ehrsson, 2017b***; ***Shergill et al., 2003***; ***Wolpe et al., 2018***). Instead, a bimanual sensorimotor context is needed such as during bimanual object manipulation and bimanual contact of the hands (***Blakemore et al., 1998b***). In the *no-contact* condition, the efference copy and the sensory input are inconsistent with the sensorimotor context of touching oneself; in this case, the context of applying force on one's finger with another finger. In other words, the attenuation model suggests that not every voluntary movement will attenuate upcoming sensory input but a naturalistic sensorimotor mapping between the movement and the sensory input is needed. Based on this, neither attenuation nor enhancement should be observed in the *no-contact* condition (i.e. no change in PSEs from *baseline*). In contrast, according to the enhancement model, if action enhances the received sensation (***Thomas et al., 2022***), then higher PSEs in the *no-contact* condition compared to the *baseline* condition should be observed (***Figure 1C***). Finally, since the enhancement model proposes that attenuation effects are attributed to unspecific (nonpredictive) gating effects caused by the simultaneous tactile stimulation of the two hands, touch in the *contact* condition should also be perceived as weaker, albeit not due to action prediction. In summary, Experiment 1 explicitly assessed whether action enhances the received touch when the index fingers of the two hands do not make contact.

The results showed a robust attenuation of the perceived touch when the two fingers made contact (*contact* condition) (***Figure 1D, E and F***): the PSEs were significantly lower in the *contact* condition than in the *baseline* condition ($W = 422.00$, $p<0.001$, $rrb = 0.82$, $CI^{95} = [0.08, 0.25]$). Similarly, the PSEs were significantly lower in the *contact* condition than in the *no-contact* condition ($W = 441.00$, $p<0.001$, $rrb = 0.90$, $CI^{95} = [0.13, 0.26]$) (***Figure 1D, E and G***). Critically, however, in the comparison that contrasts the hypotheses of the two models, the *no-contact* condition did not produce any significant change in the perceived magnitude of the touch compared to the *baseline* condition ($W = 187.00$, $p=0.360$, $rrb = -0.20$, $CI^{95} = [-0.08, 0.030]$) (***Figure 1D, E and H***). A Bayesian factor analysis

provided moderate support for the absence of any difference ($BF_{01}$ = 3.58). We also extracted the just noticeable difference (JND) from the psychophysical fits, which represents the participants' force discrimination capacity between conditions. In Experiment 1, the JNDs did not significantly differ between any of the conditions (all p>0.05).

In summary, we used both frequentist and Bayesian statistics and found no evidence that the action of the right index finger produces an enhancement of the touch received on the left index finger when the fingers did not make contact, relative to the *baseline*. Thus, Experiment 1 does not support the hypothesis of the enhancement model that action enhances predicted touch but supports the hypothesis of the attenuation model.

## Experiment 2. Previous 'enhancement' effects are driven by the baseline used

Having found no evidence for somatosensory enhancement in Experiment 1, Experiment 2 aimed to understand the potential source of the previously reported enhancement effects (*Thomas et al., 2022*). One critical methodological difference between Experiment 1 and previous evidence for enhancement concerns the *baseline* condition, as *Thomas et al., 2022* compared the participants' perception of the *no-contact* condition to a condition where the participants prepared their right index movement but received a NOGO cue to inhibit the movement, and it was that comparison that revealed enhancement effects. However, motor inhibition (i.e. planning the hand movement but not executing it) can lead to suppression of somatosensory input both on the hand that is planned to move (*Hoshiyama and Sheean, 1998*; *Voss et al., 2008*; *Walsh and Haggard, 2007*) and the hand that would receive touch if the movement was executed (*Kilteni et al., 2018*); moreover, such conditions result in a competition for attentional resources to inhibit the motor response with processes that encode sensory stimuli into memory (*Chiu and Egner, 2015a*; *Yebra et al., 2019*) (e.g. the *test* force in this paradigm). Therefore, if the motor inhibition condition used by *Thomas et al., 2022* results in a suppression of the perceived touch, a comparison of the *no-contact* condition with such a baseline would produce an apparent enhancement effect.

Thirty new naïve participants participated in Experiment 2, which included a block of *contact* trials and a block of *no-contact* trials in which participants moved their right index finger, but each block also contained randomly intermixed NOGO trials where participants were cued to withhold their movement, identical to the design of *Thomas et al., 2022* (*Figure 2A*, *Figure 2—figure supplement 1*). Participants were trained to make similar movements with their right index finger in the *contact* and *no-contact* trials. An externally generated touch condition was also included as an action-free *baseline* condition (i.e. no action execution or inhibition). If the previously reported enhancement effects are due to the baseline used, attenuated perception of touch during the motor inhibition condition (i.e. NOGO trials) should be found compared to the *baseline*, which would then lead to apparent enhancement effects upon comparison with the *no-contact* trials (*Figure 2B and C*) (**Methods**).

This hypothesis (*Figure 2B*) was confirmed. First, all the effects of Experiment 1 were replicated in the new sample (*Figure 2D, E and F & H*). The *contact* trials yielded significant attenuation (i.e. lower PSEs) compared to the *baseline* condition (t(29) = 6.79, p<0.001, d = 1.24, $CI^{95}$ = [0.18, 0.33]) (*Figure 2E & H*). Once again, there was no enhancement in the *no-contact* trials compared to the *baseline* condition (t(29) = 0.45, p=0.658, d = 0.08, $CI^{95}$ = [–0.05, 0.07]) (*Figure 2D & F*)**,** and the Bayesian analysis again yielded moderate evidence for the absence of any effects ($BF_{01}$ = 4.69). Importantly, the PSEs in the *NOGO* (motor inhibition) trials were significantly lower than the *baseline* condition both for the *contact NOGO* (t(29) = 2.99, p=0.006, d = 0.55, $CI^{95}$ = [0.02, 0.09]) and the *no-contact NOGO* trials (t(29) = 4.44, p<0.001, d = 0.81, $CI^{95}$ = [0.05, 0.13]) (*Figure 2D and E & G*). This demonstrates that NOGO trials resulted in a suppression of perceived touch on the left hand. Critically, this led to an apparent increase in the PSE of the *no-contact* trials compared to the NOGO trials in the no-contact block (t(29) = –2.98, p=0.006, d = –0.54, $CI^{95}$ = [–0.12,–0.02]), mimicking an 'enhancement' effect. Finally, PSEs were significantly lower in the *contact* trials than in the NOGO trials (t(29) = 5.91, p<0.001, d = 1.08, $CI^{95}$ = [0.13, 0.27]), while the NOGO trials in the *contact* and *no-contact* blocks did not significantly differ (t(29) = 1.58, p=0.126, d = 0.29, $CI^{95}$ = [–0.01, 0.08], $BF_{01}$ = 1.697). In Experiment 2, the JNDs were not significantly different between conditions (p>0.05), except in the *contact* condition, where the JNDs were significantly higher than in the *baseline* condition (t(29) = –2.52, p=0.017, d = –0.46, $CI^{95}$ = [–0.06,–0.01]).

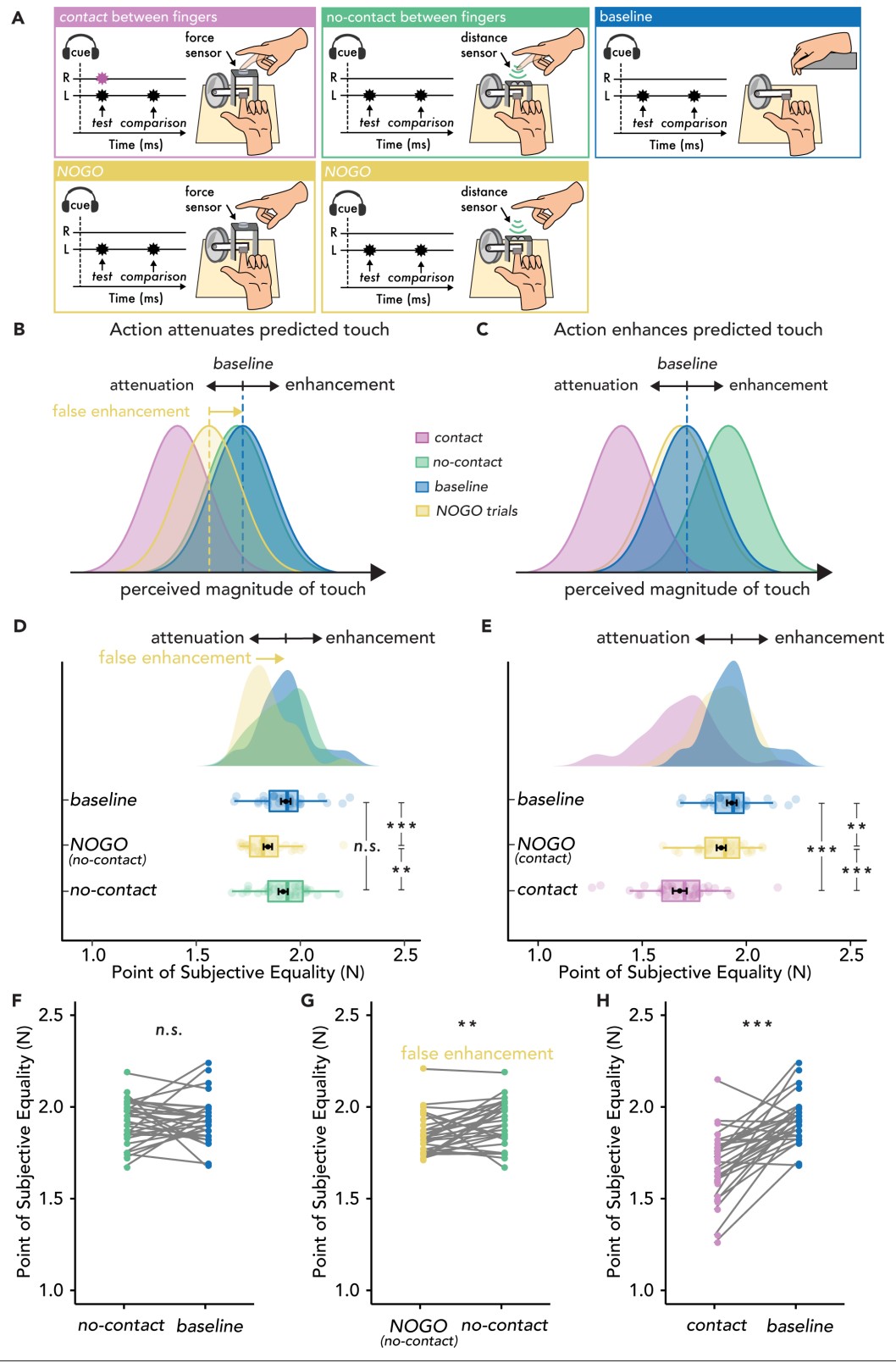

**Figure 2.** Experimental methods, hypotheses, and results of Experiment 2 (n=30). (**A**) The *contact* (magenta) and *no-contact* (green) trials were identical to those of Experiment 1, with the only difference that in 50% of the trials, the participants had to inhibit their movement (*NOGO* trials - yellow), and the test force was delivered automatically. The *baseline* condition (blue) was identical to that of Experiment 1. (**B**) Hypotheses based on

*Figure 2 continued on next page*

*Figure 2 continued*

the attenuation model. Touch in the *contact* condition (magenta) should be perceived as weaker than in the *baseline* (blue), but touch in the *no-contact* condition (green) should be perceived similarly to that in the *baseline* (blue). Critically, touch may also be perceived as weaker than *baseline* in the *NOGO* trials, resulting in a 'false enhancement' of the *no-contact* trials. (**C**) Hypotheses based on the enhancement model. Touch in the *no-contact* condition (green) should be perceived as stronger than in the *baseline* (i.e. enhanced) (blue), but touch in the *contact* condition (magenta) should be perceived as weaker than the *baseline* (blue). (**D–H**) Data are color-coded per condition. (**D–E**) Box plots show the median and interquartile ranges for the point of subjective equality (PSE) values in the *baseline* and *no-contact* blocks (**D**) and in the *baseline* and *contact* blocks (**E**). Black circles and error bars show the mean PSE ± SEM, and the raincloud plots show the individual PSE values and their distributions. (**F–H**) Line plots for individual participant PSE values illustrating no significant differences in the PSE values between *no-contact* and *baseline* (**F**), significantly higher PSEs for the *no-contact* versus *NOGO* trials in the same block (**G**) and significantly lower PSEs in the *contact* versus *baseline* trials (**H**). (**p<0.01, ***p<0.001, n.s. non-significant).

The online version of this article includes the following figure supplement(s) for figure 2:

**Figure supplement 1.** Fitted logistic models based on the participants' responses under each condition of Experiment 2.

In summary, identical to Experiment 1, we used both frequentist and Bayesian statistics and did not find any evidence that the action of the right index finger produces an enhancement of the touch received on the left index finger when the fingers do not make contact, relative to the *baseline*. Moreover, we showed that the purported 'enhancement' effect (*Thomas et al., 2022*) is driven by a suppression of the perceived intensity of touch following a cue to inhibit the planned movement, both relative to the *baseline* and to the condition where the fingers do not make contact (*no-contact*). Thus, Experiment 2 does not support the hypothesis that action enhances predicted touch.

## Experiment 3. Action attenuates predicted touch, even without simultaneous stimulation of the active hand

Within the framework of internal forward models, the absence of attenuation in the *no-contact* conditions of Experiments 1 and 2 is expected, given the lack of a sensorimotor context conducive to perceiving the touch as self-generated: when contact is never made, the forces applied on the passive left hand are only arbitrarily, and not causally, associated with the movement of the active right hand. However, if there is an expectation of contact between the hands, regardless of whether the contact is eventually made, touch should be attenuated as previously shown (*Bays et al., 2006*). Alternatively, it could be argued that the absence of attenuation in the *no-contact* conditions is caused by the lack of simultaneous tactile stimulation of the active hand rather than by predictive mechanisms.

In Experiment 3, this hypothesis was explicitly tested with thirty new naïve participants (*Figure 3—figure supplement 1*). The same *contact* and *no-contact* conditions were included as in Experiment 1, but their relative frequency was manipulated within the same block (*contact* trials: 80%, *no-contact* trials: 20%) (*Figure 3A*). The force sensor the participants tapped in the *contact* trials was now attached to a platform that could be automatically retracted. In the *contact* trials, participants tapped the force sensor to trigger the *test* force identically to Experiments 1 and 2, but in the *no-contact* trials, the platform automatically retracted before trial onset, unbeknownst to the participants, leading them to unexpectedly miss the contact with the sensor but still trigger the *test* force only by the position of their right index finger. Participants' vision was occluded so that they could not know whether the next trial would be a *contact* or a *no-contact* trial. According to the attenuation model, providing a bimanual sensorimotor context in 80% of the trials should lead participants to form predictions about the somatosensory consequences of their movement in most of the trials and thus attenuate the received touch on their passive left hand compared to the *baseline*, even if the touch of their active hand was unexpectedly missed (*Figure 3B*). In contrast, if attenuation is a nonpredictive process caused by the mere simultaneous tactile stimulation of the active finger, no attenuation effects in the *no-contact* trials should be observed with respect to the *baseline* (*Figure 3C*).

The results showed a robust attenuation in both *contact* and *no-contact* trials with respect to the *baseline* condition, regardless of whether contact was made (*Figure 3D, E and F & H*): the PSEs were significantly lower in the *contact* condition than in the *baseline* condition ($t(29) = 8.06$, p<0.001, $d = 1.47$, $CI^{95} = [0.16, 0.27]$) and lower in the *no-contact* condition than in the *baseline*

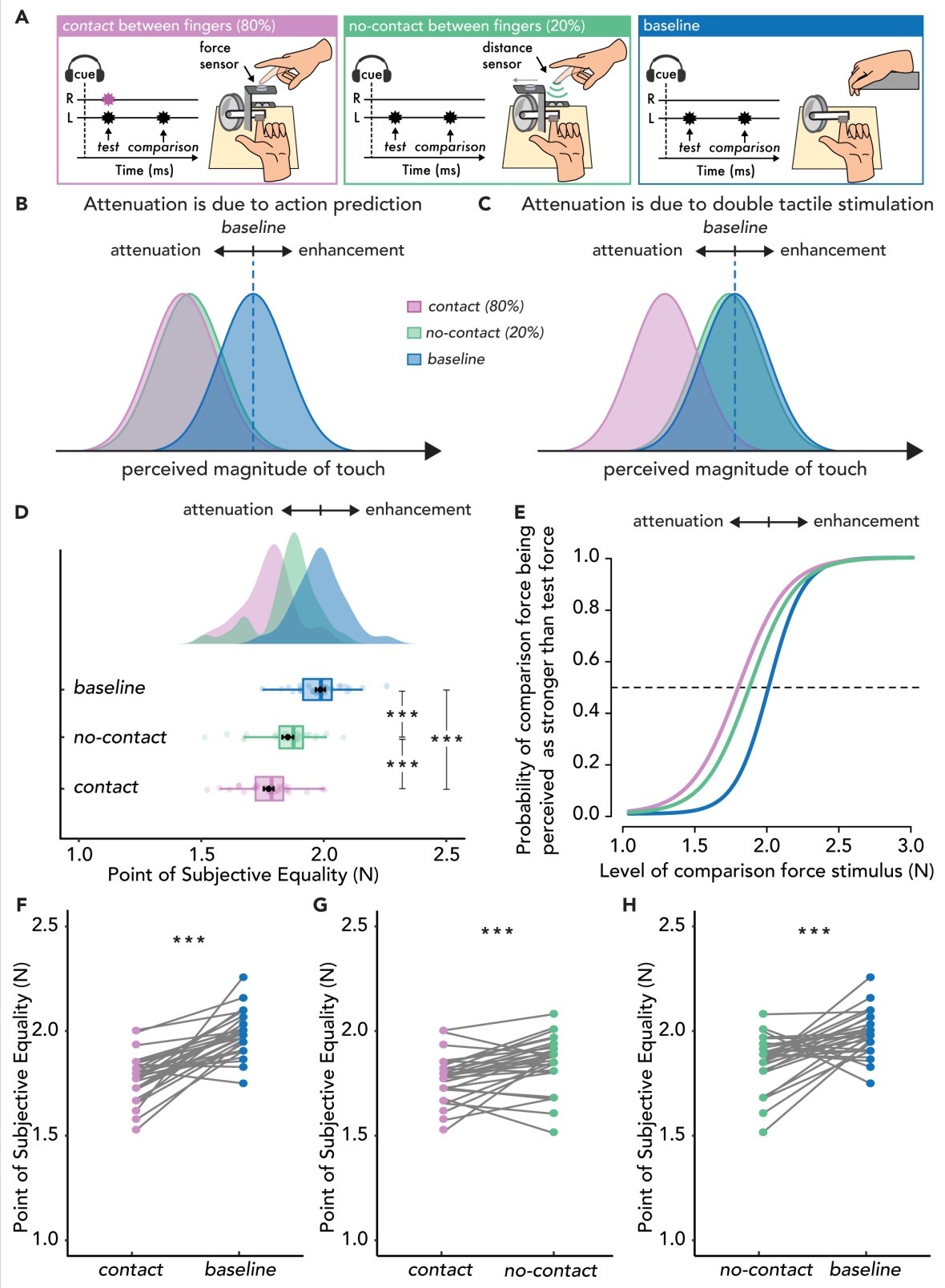

**Figure 3.** Experimental methods, hypotheses, and results of Experiment 3 (n=30). (**A**) The *contact* (magenta) and *no-contact* (green) conditions were identical to those of Experiment 1, with the only difference being their relative proportion (*contact* trials 80%, *no-contact* trials 20%). In the *no-contact* trials, the force sensor was automatically retracted, unbeknownst to the participant, revealing the distance sensor placed below. The *baseline* condition (blue) was identical to that of Experiments 1 and 2. (**B**) Hypotheses based on the attenuation model. If attenuation is due to action prediction, then the

*Figure 3 continued*

perceived magnitude of touch should be reduced in both the *contact* (magenta) and *no-contact* conditions (green) compared to the baseline (blue). (**C**) Hypotheses based on the enhancement model. If attenuation effects are driven by simultaneous touch on the active hand, then the perceived magnitude of touch should be reduced only in the *contact* condition (magenta) compared to the *baseline* (blue) and not in the *no-contact* condition (green) which should be similar to the *baseline* (blue). (**D–H**) Data are color-coded per condition. (**D**) Box plots show the median and interquartile ranges for the point of subjective equality (PSE) values per condition, black circles and error bars show the mean PSE± SEM, and the raincloud plots show the individual PSE values and their distributions. (**E**) Group psychometric functions for each condition. The leftward shift of the curves in the *contact* and *no-contact* conditions indicates attenuated PSE values compared to the *baseline*. (**F**) Line plots for individual participant PSE values illustrate significantly lower PSEs for the *contact* versus *baseline* condition, (**G**) significantly lower PSEs in the *contact* versus *no-contact* condition and (**H**) significantly lower PSEs in the *no-contact* versus *baseline*. (***p<0.001).

The online version of this article includes the following figure supplement(s) for figure 3:

**Figure supplement 1.** Fitted logistic models based on the participants' responses under each condition of Experiment 3.

**Figure supplement 2.** Distance between the fingers (*contact* condition minus *no-contact* condition) for each experiment at the time of the test force for each experiment (n=30).

**Figure supplement 3.** Forces exerted on the left index finger (Test force) and on the force sensor by the right index finger (Peak active force) (n=30).

condition ($t(29)$ = 4.45, p<0.001, $d$ = 0.81, $CI^{95}$ = [0.07, 0.20]). The magnitude of attenuation was larger in the *contact* condition than in the *no-contact* condition, with significantly lower PSEs in the *contact* condition than in the *no-contact* condition ($t(29)$ = 3.94, p<0.001, $d$ = 0.72, $CI^{95}$ = [0.04, 0.12]) (*Figure 3G*). The difference in the magnitude of attenuation was modest ($\cong$ 35%). These findings indicate that the perceived intensity of touch on the passive left hand was significantly attenuated both when the active hand received touch or not, thereby ruling out the possibility that attenuation is merely due to simultaneous tactile stimulation (*Press et al., 2023*; *Thomas et al., 2022*).

The results of Experiment 3 emphasize the predictive nature of somatosensory attenuation, which is observed only when the sensorimotor context allows the formation of such predictions. In Experiments 1 and 2 attenuation was only observed in the *contact* conditions, whereas attenuation was also observed in the *no-contact* condition of Experiment 3, where participants expected that they would make contact (i.e. the sensorimotor context was conducive to predicting self-touch). Thus, larger differences between the PSEs in the *contact* and *no-contact* conditions were observed for Experiments 1 and 2 compared to Experiment 3. To further strengthen our interpretation, we performed an ANOVA on the difference in the PSEs between the *contact* and *no-contact* conditions across all three experiments. The ANOVA revealed a significant main effect of Experiment ($F(2, 87)$ = 8.05, p<0.001, $\eta_p^2$ = 0.156), with Bonferroni corrected post hoc comparisons indicating significant differences between Experiments 1 and 3 ($t(58)$ = 3.10, p=0.008, $d$ = 0.80, $CI^{95}$ = [0.03, 0.23]) as well as Experiments 2 and 3 ($t(58)$ = 3.76, p<0.001, $d$ = 0.97, $CI^{95}$ = [0.06, 0.26]), but no significant differences between Experiments 1 and 2 ($t(58)$ = –0.66, p=1.000, $d$ = –0.17, $CI^{95}$ = [–0.13, 0.07], $BF_{01}$ = 3.308). Similarly, an ANOVA on the difference in the PSEs between the *baseline* and *no-contact* conditions across all three experiments revealed a significant main effect of Experiment ($F(2, 87)$ = 7.59, p<0.001, $\eta_p^2$ = 0.149). Bonferroni corrected post hoc comparisons indicated significant differences between Experiments 1 and 3 ($t(58)$ = –3.76, p<0.001, $d$ = –0.97, $CI^{95}$ = [-0.27,–0.06]) and between Experiments 2 and 3 ($t(58)$ = –2.76, p=0.021, $d$ = –0.71, $CI^{95}$ = [–0.22,–0.02]), but no significant difference between Experiments 1 and 2 ($t(58)$ = –0.99, p=0.963, $d$ = –0.26, $CI^{95}$ = [–0.16, 0.06], $BF_{01}$ = 2.537).

Thus, *no-contact* trials elicited attenuated perceptions only when the sensorimotor context allowed for a prediction of self-touch (Experiment 3) and not when the action was only arbitrarily associated with the touch (Experiments 1 and 2).

Regarding the JNDs, the values were significantly lower in the *baseline* condition than in the *contact* condition ($t(29)$ = –3.61, p=0.001, $d$ = –0.66, $CI^{95}$ = [–0.09,–0.02]) and the *no-contact* condition ($t(29)$ = –2.77, p=0.010, $d$ = –0.51, $CI^{95}$ = [–0.08,–0.01]), but there were no significant differences in the JNDs between the *contact* and *no-contact* conditions ($t(29)$ = –0.65, p=0.521, $d$ = –0.12, $CI^{95}$ = [–0.02, 0.04], $BF_{01}$ = 4.233). Thus, in contrast to the PSEs, there were no significant differences in the JNDs between the *contact* and *no-contact* conditions in any of the three experiments, demonstrating that the two conditions yielded the same discrimination capacity.

## Kinematic results

To ensure that the movements in both the *contact* and *no-contact* conditions were comparable, participants were trained to make the same movement with their right index finger between two visual markers positioned 5 cm apart in Experiments 1 and 2. This training was not required for Experiment 3, in which participants were blindfolded and unaware that the distance sensor had been retracted on *no-contact* trials.

Given that the right index finger movement ends on impact with the force sensor in the *contact* condition but not in the *no-contact* condition, the kinematics of the finger movements are likely to differ between conditions. The active finger was slightly closer to the passive finger at the time of the *test* force in the *contact* compared to the *no-contact* condition in Experiments 1 and 2 (by less than 5 mm). Specifically, in Experiment 1, the position of the right index finger at the time of the *test* force was marginally closer to the left index finger in the *contact* condition than in the *no-contact* condition (mean difference = 2.32 mm, SEM = 1.14: $t(29)$ = –2.04, p=0.051, $d$ = 0.37, $CI^{95}$ = [–0.74, 0.01], $BF_{01}$ = 0.845). In Experiment 2, the position of the right index finger at the time of the *test* force was slightly closer to the left index finger in the *contact* condition than in the *no-contact* condition (mean difference = 4 mm, SEM = 1.8: $t(29)$ = –2.29, p=0.030, $d$ = 0.42, $CI^{95}$ = [–0.79,–0.04]). In Experiment 3, participants were unaware that they would miss the force sensor in the *no-contact* trials, so with no force sensor to stop the movement, their right index finger continued further downwards. Here, the position of the right index finger at the time of the *test* force was closer to the left index finger in the *no-contact* condition than in the *contact* condition (mean difference = 25.7 mm, SEM = 2.0: $t(29)$ = 12.79, p<0.001, $d$ = 2.34, $CI^{95}$ = [1.63, 3.03]). Critically, the closer position of the right index finger to the left index finger in the *no-contact* trials compared to the *contact* trials cannot explain the attenuation findings in the *no-contact* condition since the attenuation in the *contact* trials was stronger compared to that in the *no-contact* trials (***Figure 3—figure supplement 2***).

It could be argued that our differences between the *contact* and *no-contact* conditions across experiments are driven by differences in the timing of the two setups. This explanation is unlikely since we observed robust attenuation in the *no-contact* condition of Experiment 3 but not in Experiments 1 and 2 while the setups remained the same. To further control that the attenuation observed in the *no-contact* condition of Experiment 3 but not in the *no-contact* conditions of Experiments 1 and 2 was not driven by timing differences in the participants' movement between experiments, we calculated the time the *test* force was delivered in the *contact* and *no-contact* conditions with respect to their movement endpoint (i.e. the force sensor press in the *contact* condition and the lowest vertical position in the *no-contact* condition). On average, the *test* force was applied 24 ms earlier in the *no-contact* compared to the *contact* condition in Experiment 1 (SEM = 4.20), 11 ms earlier in the *no-contact* compared to the *contact* condition in Experiment 2 (SEM = 4.53) and 15 ms earlier in the *no-contact* compared to the *contact* condition in Experiment 3 (SEM = 2.67). An ANOVA revealed a significant main effect of Experiment ($F(2, 87)$ = 3.32, p=0.041, $\eta_p^2$ = 0.071), with Bonferroni corrected post hoc comparisons indicating significant differences between Experiment 1 and Experiment 2 ($t(58)$ = 2.50, p=0.043, $d$ = 0.65, $CI^{95}$ = [0.62, 26.83], $BF_{01}$ = 0.500). Importantly, there were no significant differences between neither Experiments 1 and 3 ($t(58)$ = 1.80, p=0.226, $d$ = 0.47, $CI^{95}$ = [–3.21, 22.99], $BF_{01}$ = 0.744), nor Experiments 2 and 3 ($t(58)$ = –0.69, p=1.000, $d$ = –0.18, $CI^{95}$ = [–16.93, 9.27], $BF_{01}$ = 3.051) demonstrating that the attenuation observed in Experiment 3 cannot be attributed to time differences between the participants' movement and the received touch.

## Discussion

In the present study, two opposing hypotheses regarding how action influences somatosensory perception were contrasted: the attenuation model and the enhancement model. Our findings demonstrate that action does not enhance (Experiments 1 and 2) but attenuates the predicted touch (Experiment 3) compared to identical touch in the absence of action. Before discussing the findings, it is important to emphasize that to draw conclusions about whether perception is attenuated or enhanced in an experimental condition including action, it is necessary to include a baseline condition without action. Only comparing conditions that include action prevents differentiating a genuine effect of enhancement (or attenuation) from an effect of reduced attenuation (or enhancement) in one of the two conditions. This also applies to experimental manipulations that contrast predicted with

unpredicted somatosensory stimuli during action (see *Thomas et al., 2022* for such comparisons). To this end, a *baseline* condition of pure somatosensory exafference (i.e. externally generated touch) was included in all the present experiments that allowed us to distinguish the direction of the effects.

The results of Experiment 1 showed a robust attenuation of the touch applied to the passive left hand when the two hands made contact (*contact* condition), but neither attenuation nor enhancement was caused by the mere movement of the right hand (*no-contact* condition). In contrast, the perceived intensity of touch in the *no-contact* condition was similar to that of the *baseline* (i.e. externally generated touch). These findings are in line with the results of *Bays et al., 2006*, who found no change in the participants' somatosensory perception when the two hands did not make contact, but do not replicate those of *Thomas et al., 2022*, who observed enhancement of touch in a *no-contact* condition. Experiment 2 further investigated the source of the previously reported enhancement effects and showed that they are in fact driven by the reference condition used: enhancement was observed only relative to a condition in which the touch was applied rapidly following a cue to inhibit the movement (i.e. a 'do not move' cue), as in previous research (*Thomas et al., 2022*), but not relative to an externally generated touch condition (our baseline). In support of this claim, Experiment 2 showed that a cue to inhibit the movement results in a significant reduction in the perceived intensity of the imperative stimulus (i.e. touch on the passive hand) compared to baseline perception. This is in line with previous evidence showing reduced amplitudes of somatosensory evoked potentials for tactile stimuli presented shortly following a cue to inhibit a movement (*Hoshiyama and Sheean, 1998*), reduced perceived amplitude for tactile stimuli under the mere expectation to move (*Voss et al., 2008*; *Voss et al., 2006*), and reduced encoding of sensory stimuli following motor inhibition (*Chiu and Egner, 2015b*; *Yebra et al., 2019*). Therefore, rather than participants' perception being enhanced in the *no-contact* condition, it is a reduction of the perceived intensity following the cue to inhibit the movement that leads to an apparent enhancement. In contrast, by including a novel externally generated touch condition as a *baseline* that involved neither motor planning nor response inhibition, it became clear that there were no enhancement effects.

Experiment 3 demonstrated that action attenuates the predicted touch even if the active hand does not receive simultaneous tactile stimulation with the passive hand. When participants simply moved their right hand to trigger the touch on their left hand (*no-contact* trials in Experiments 1 and 2), no change in their somatosensory perception was found. This suggests that an arbitrary mapping between the movement of the right hand and the delivery of touch on the left hand is insufficient to elicit attenuation. In contrast, when participants expected that they would touch their own hand (Experiment 3), significant attenuation was observed, even when the active hand unexpectedly missed the contact (*no-contact* trials, 20%). This pattern of results can be explained by the attenuation model, which suggests that the self-generation of a stimulus is in itself not sufficient for predictive attenuation, but that the action and its predicted sensory feedback are consistent with touching one's own body (*Bays and Wolpert, 2008*). Thus, a sensorimotor context that closely resembles tapping directly on the left index finger with the right index finger (self-touch) was critical for predictive attenuation (*Bays et al., 2006*). The findings of Experiment 3 also contradict the suggestion that attenuation on a passive hand reflects a nonpredictive gating of tactile input during movement of the active hand (*Press et al., 2023*; *Thomas et al., 2022*). Indeed, several earlier studies showed gating effects only on the moving limb (movement effector) and not on the passive limb (*Chapman et al., 1987*; *Cohen and Starr, 1987*; *Colino et al., 2014*; *Papakostopoulos et al., 1975*; *Pertovaara et al., 1992*; *Rushton et al., 1981*), and we have also recently shown that experimental paradigms identical to the one used in the present study produce attenuation effects but not gating effects on the passive hand (*Kilteni and Ehrsson, 2022*). Overall, the results suggest that attenuation effects are driven by action prediction and not the simultaneous tactile stimulation of the two hands. From an ecological point of view, in every self-touch behavior, we necessarily receive somatosensory input on the active hand and the body part that passively receives touch ('touchant-touché' [*Schütz-Bosbach et al., 2009*]), and it is within these sensorimotor contexts that the brain forms predictions about the somatosensory consequences on multiple body parts (*Bays and Wolpert, 2008*).

Interestingly, the magnitude of attenuation was greater in the *contact* compared to the *no-contact* condition by approximately 35% in Experiment 3. This could suggest that the attenuation effect observed in the *contact* condition arises from multiple sources, one being the motor-based prediction and another source being the simultaneous touch on the active hand. However, this interpretation is

inconsistent with the findings that somatosensory attenuation has been previously observed in the absence of simultaneous tactile stimulation, for example, when imagining but not executing the right hand movement (*Kilteni et al., 2018*) or just before the hands make contact (*Bays et al., 2005*). Similarly, no somatosensory attenuation is observed in the absence of action prediction, even if the two hands receive simultaneous tactile stimulation; for example, the passive displacement of the right hand towards the left hand that is accompanied by a double touch (*Kilteni et al., 2020*) or the mere delivery of simultaneous tactile stimulation on both hands (*Bays et al., 2005*) does not produce attenuation. Instead, our finding that the magnitude of attenuation was greater in the *contact* than the *no-contact* condition in Experiment 3 may instead reflect a decrease of attenuation in the *no-contact* trials, due to the unexpected omission of contact influencing the perceived magnitude of touch in a postdictive manner. Specifically, the unexpected omission of contact in the minority of trials (20%) could be seen as a form of stimulus omission akin to so-called 'silent oddballs' that are known to generate prediction errors (*Busse and Woldorff, 2003*; *Karamürsel and Bullock, 2000*; *SanMiguel et al., 2013a*; *SanMiguel et al., 2013b*). That is, although participants clearly attenuated the touch predicted by their movement in the *no-contact* trials, their expectation of contact was necessarily violated. In one of our previous studies (*Kilteni et al., 2019*) using the same force discrimination task as in the present study we showed in three experiments that participants experienced the touch to be stronger (i.e. less attenuation) and more ticklish when triggering a prediction error (<17% frequency). Similarly, research outside the domain of action (*Kusnir et al., 2020*) has found that perception is improved for tactile events delivered at unpredictable compared to predictable temporal onsets, also suggesting that tactile perception is facilitated in the presence of prediction error. In line with these previous results, the decreased attenuation we observed in the *no-contact* condition could be attributed to the prediction error. Additionally, this violation could be considered a novel event given its infrequency. Novel events can have several consequences for cognition, including transient enhancements of perception (*Schomaker and Meeter, 2012*), facilitated encoding of information into working memory (*Mayer et al., 2011*) as well as changes in the allocation of attentional resources in a postdictive manner (for a review of the effects of novelty on cognition see *Schomaker and Meeter, 2015*).

Some authors have criticized the comparison of perceived intensity of sensory stimuli in self-generated and externally generated conditions (*Press et al., 2020*; *Yon et al., 2018*; *Yon et al., 2021*). A frequent criticism is that tactile input may be 'predicted' in self-generated conditions through action and 'unpredicted' in externally generated (passive) conditions that do not involve action. This concern can be ruled out since in all three experiments, the stimulus in the *baseline* (i.e. externally generated touch) condition was delivered at a fixed timepoint (800 ms after the cue); therefore, participants could predict it in the absence of motor-based predictions. Second, it has been argued that somatosensory attenuation findings may be driven by dual-task requirements (*Press et al., 2020*) present only in self-generated conditions that could increase the working memory load or result in a shift of attention towards the active hand. A similar criticism has been suggested in which the mere movement of the active hand in self-generated conditions can result in a nonpredictive gating of tactile input. However, these explanations can also be ruled out, given that dual-task requirements and movement of the active hand were present in the *contact* and *no-contact* conditions of all three experiments without concomitant attenuation effects. Indeed, if dual-task requirements, or the mere presence of movement during the perceptual task, were sufficient to cause attenuation effects, then attenuation would have been observed in the *no-contact* conditions of Experiments 1 and 2. Furthermore, if the mere simultaneous tactile stimulation on the active hand was responsible for the attenuation effect, we would not have observed attenuation in the *no-contact* condition of Experiment 3. Finally, alternative explanations based on differences in other psychophysical parameters, movement kinematics, or timings between experiments can also be ruled out.

How can the findings that action prediction attenuates touch be reconciled with those showing that expectations outside the domain of action improve sensory perception (*Press et al., 2020*)? While it is difficult to directly compare these lines of research because of differences in the sensory modality investigated, the task designs used, and the perceptual measures employed, we speculate that there are numerous possible reasons why action prediction may not have the same effect on perception as prediction mechanisms outside the domain of action. First, research on action-based predictions concerns ubiquitous associations between actions and their sensory consequences that we are continually exposed to throughout the lifespan. For example, we are constantly exposed to

associations between our motor behaviors and their tactile consequences during self-touch, even as early as 13 weeks in utero (*Kurjak et al., 2003*). In contrast, research on sensory expectations outside the domain of action primarily concerns arbitrary associations between stimuli that are typically learned only during the time course of a given task. It is, therefore, conceivable that separable mechanisms may operate to predict action effects versus stimulus-stimulus associations (see *Dogge et al., 2019a*; *Dogge et al., 2019b* for discussion). Second, higher-level expectations, such as explicit prior knowledge that a specific sensory event is likely, might not operate in the same way as lower-level action predictions; for example, it has been proposed that action-based predictions inhibit expected stimuli, while sensory expectations potentiate the expected sensory inputs (*de Lange et al., 2018*). Most importantly, from a theoretical perspective, attenuating the predicted sensory consequences of actions does not necessarily mean that the brain forms inaccurate representations of the world but instead indicates a flexible strategy that prioritizes more informative externally generated events.

It is important to emphasize that this study investigated the perception of self-generated touch during an action directed toward one's own body. Thus, perception was assessed on a passive body part that was the target of a movement of an active body part (i.e. moving the right hand to touch the static left hand). This focus was chosen because extensive previous research has shown deteriorated performance when detecting or discriminating touch (both self-generated and externally generated) delivered on a moving limb, a phenomenon referred to as movement-related *gating* (*Chapman et al., 1996*; *Chapman et al., 1987*; *Voudouris and Fiehler, 2021*; *Voudouris and Fiehler, 2017*; *Williams et al., 1998*). By assessing somatosensory perception on a static limb, we were able to isolate the effect of action prediction on somatosensory perception from other perceptual phenomena related to the movement per se including movement-related gating (*Chapman et al., 1987*; *Cohen and Starr, 1987*; *Colino et al., 2014*; *Kilteni and Ehrsson, 2022*; *Papakostopoulos et al., 1975*; *Pertovaara et al., 1992*; *Rushton et al., 1981*). However, it is interesting to note that during the active exploration of objects through touch, referred to as *active touch*, self-generated touch also occurs, albeit only on the moving limb. Studies on active touch have observed improved perception of object features, such as texture or edge orientation when actively exploring an object (*Morley et al., 1983*; *Pruszynski et al., 2018*) compared to passive exposure (*Bensmaia et al., 2008*; *Peters et al., 2015*). At first, the improved perception observed during active touch might appear paradoxical to observations of movement-related gating and predictive attenuation effects: if anything, active touch should result in worse perception than passive touch. However, it has been shown that humans optimize their exploratory movements to minimize these effects and maximize the quality of their sensory feedback. For example, it is known that the degree of gating decreases as movement speed slows (*Angel and Malenka, 1982*; *Chapman et al., 1996*; *Schmidt et al., 1990*) and it has been proposed that participants optimize the speed of movement during active touch to avoid high speeds associated with tactile gating (*Cybulska-Klosowicz et al., 2011*). Similarly, when exploring objects through active touch, weaker forces are typically applied (e.g. <1 N [*Olczak et al., 2018*; *Tanaka et al., 2014*]) that elicit less attenuation than stronger forces, as shown in the classic force-matching task (*Shergill et al., 2003*). Thus, differences in the task context might primarily influence how one moves (e.g. the act of swatting a fly from one's arm versus feeling for a bump on one's skin), which can have disparate ensuing effects on perception depending on the movement kinematics.

Debates between researchers supporting attenuation or enhancement are useful and fruitful for scientific dialogue and advancement. The present study revisited recent findings on somatosensory enhancement during action and showed that when the requirement to inhibit the action is removed from the baseline, action predictions do not enhance but attenuate the received somatosensory input. Our results are in strong alignment with animal studies showing that action attenuates the predicted sensory consequences (for reviews, see [*Brooks and Cullen, 2019*; *Crapse and Sommer, 2008*; *Cullen, 2004*; *Schneider and Mooney, 2018*; *Straka et al., 2018*]). For example, crickets and mice suppress auditory reafferent signals but maintain their sensitivity to external sounds (*Audette et al., 2022*; *Poulet and Hedwig, 2003*; *Poulet and Hedwig, 2006*; *Schneider et al., 2018*), the weakly electric fish attenuates its electrosensory reafference to respond to externally generated electrical signals (*Cullen, 2004*; *Sawtell, 2017*), and primates attenuate their vestibular reafference and activate vestibular-related reflexes only when the vestibular input is exafferent (*Brooks et al., 2015*; *Cullen, 2012*; *Roy and Cullen, 2004*). To this end, the results of the present study prompt a reappraisal of

recent experimental findings upon which theoretical frameworks proposing a perceptual enhancement by action prediction are based.

## Methods

### Participants

Thirty naive adults participated in Experiment 1 (18 females, aged 18–36, 28 right-handed and 2 ambidextrous), thirty naive adults participated in Experiment 2 (17 females, aged 20–37, 29 right-handed and 1 left-handed) and thirty naive adults participated in Experiment 3 (12 females, aged 21–40, 28 right-handed, 1 ambidextrous, 1 left-handed). Current or history of psychological or neurological conditions, as well as the use of any psychoactive drugs or medication to treat such conditions, were criteria for exclusion. The sample size of each Experiment was decided prior to data collection based on our previous studies using the same task (*Kilteni et al., 2019*; *Kilteni and Ehrsson, 2022*). Handedness was assessed using the Edinburgh Handedness Inventory (*Oldfield, 1971*). All experiments were approved by the Swedish Ethical Review Authority (registration no. 2021–03790). Participants provided written informed consent and were compensated for their time.

### Pre-registration of experiments

The method and analysis plan for each experiment was pre-registered on the Open Science Framework (OSF) prior to data collection (Experiment 1 https://osf.io/9jkqt, Experiment 2 https://osf.io/hs8au, Experiment 3 https://osf.io/gkwu7).

### Modifications/additions to the pre-registered plan

The only modification from the pre-registered plan was a reduction in the number of trials administered in Experiment 3. Following initial piloting, it was deemed appropriate to reduce the overall length of the task by including fewer trials (see *Methods details*). Three additional ANOVAs were also conducted to compare the results across the three experiments and the kinematics across experiments.

### Experimental setup

Participants sat comfortably on a chair with their arms placed on a table. The left hand rested palm up, with the index finger placed on a molded support. On each trial, a motor (Maxon EC Motor EC 90 flat; Switzerland) delivered two forces (the *test* force and the *comparison* force) on the pulp of the left index finger through a cylindrical probe (25 mm height) with a flat aluminum surface (20 mm diameter) attached to a lever on the motor. A force sensor (FSG15N1A, Honeywell Inc; diameter, 5 mm; minimum resolution, 0.01 N; response time, 1 ms; measurement range, 0–15 N) within the probe recorded the forces applied on the left index finger. Following the presentation of the two forces, participants were asked to verbally report which of the two forces felt stronger – the first or the second. A second identical force sensor within an identical cylindrical probe ('active force sensor') was placed on top of, but not in contact with, the probe of the left index finger.

### Force-discrimination task

Participants judged the intensity of a *test* force and a *comparison* force (100 ms duration each) separated by a random interval between 800 ms and 1200 ms in a two-alternative forced choice (2AFC) task. The intensity of the test force was 2 N, while the intensity of the comparison force was systematically varied among seven force levels (1, 1.5, 1.75, 2, 2.25, 2.5, or 3 N). In all the conditions, the forces were delivered by the same motor, to precisely control their magnitude. However, the source of the force was manipulated across conditions such that the force was triggered by the participants' contact with a force sensor (*contact* condition), their finger movement (*no-contact* condition), or automatically by the stimulus computer (*baseline* condition).

### Experimental design and procedures

#### Experiment 1

There were three conditions in Experiment 1 presented in three blocks separately: the *baseline* (i.e. externally generated touch) condition, the *contact* condition, and the *no-contact* condition.

In the *baseline* condition, participants did not move their limbs but passively received the *test* and the *comparison* force on the left index finger. This *baseline* condition was used to assess the participants' somatosensory perception in the absence of any movement (*Bays et al., 2005*; *Kilteni et al., 2020*; *Kilteni et al., 2019*). Each trial began with an auditory cue (100 ms duration, 997 Hz) followed by the *test* force delivered to the participant's left index finger 800 ms after the cue by the motor. The *comparison* force was then delivered at a random interval between 800 ms and 1200 ms after the *test* force.

In the *contact* condition, participants started each trial by holding their right index finger approximately 10cm above their left index finger. The start position was marked with a visual marker placed next to their right index finger while the final position was the probe of the active force sensor. The same auditory cue was presented as in the *baseline* condition, but participants now moved their right index finger downwards toward their left index finger and actively tapped on the active force sensor placed on top of, but not in contact with, the probe. The participant's active tap on the force sensor triggered the motor to apply the *test* force on their left index finger (threshold 0.2N). The tap of their right index finger triggered the test force on their left index finger with an intrinsic delay of 36 ms (*Figure 3—figure supplement 3*). The *comparison* force was then delivered at a random interval between 800 ms and 1200 ms after the *test* force. As in our previous studies (*Asimakidou et al., 2022*; *Kilteni et al., 2021*; *Kilteni and Ehrsson, 2022*), participants were asked to tap, neither too weakly nor too strongly, with their right index finger, 'as if tapping the screen of their smartphone'. This instruction was provided to ensure that the relationship between the force they applied with their right index finger on the force sensor and the force they received on their left index finger by the motor (2N) remained approximately constant throughout the experiment, thereby establishing perceived causality (*Bays and Wolpert, 2008*; *Kilteni, 2023*). Indeed, across all three experiments, we observed low variability in the exerted forces (*Figure 3—figure supplement 3D-F* ) (± 0.13N SEM in Experiment 1, ± 0.12N SEM in Experiment 2 and ± 0.11N SEM in Experiment 3) establishing an approximately constant gain between the exerted and the applied forces. Critically, it has been previously shown that the magnitude of attenuation remains unaffected by halving or doubling the gain between the force applied by the active finger and the force delivered on the passive hand as long as the gain remains constant (*Bays and Wolpert, 2008*). This should not be surprising given that when one finger transmits a force through an object to another finger, the resulting force depends also on the object's properties (e.g. shape, material, contact area) and the angle at which the finger contacts the object.

In the *no-contact* condition, the participants started each trial by holding their right index finger 10cm above their left index finger, identically to the *contact* condition. For this block, the active force sensor was removed and replaced by a distance sensor that detected the position of the right index finger. The distance sensor was placed on top of, but not in contact with, the probe. Following the same auditory cue, participants moved their right index finger towards their left index finger. To restrict the participants' right index finger movement within similar movement ranges as in the *contact* condition and avoid contact with the distance sensor placed underneath, a second visual marker indicated the final position of the right index finger above the distance sensor (5cm from the marker indicating the initial position). The distance sensor was connected to an Arduino microcontroller that controlled a servo motor. The servo motor and the force sensor were placed 2 meters away from the participants' hands and hidden from view. Once the distance sensor detected that the position of the right index finger became smaller than a preset threshold, it triggered the servo motor that hit the force sensor and triggered the *test* force. We accounted for the additional delay of the distance sensor by setting the position threshold for the distance sensor slightly higher than the lowest position the participants were asked to reach with their right index finger. Therefore, the test force would be delivered at a similar timing to the participants' right index finger endpoint between the *contact* and *no-contact conditions*. This position threshold was set based on significant pilot testing prior to all experiments. Indeed, there was a minimal time difference between the two setups across experiments (17 ms average delay) with the *no-contact* condition leading, rather than lagging the *contact* one (see **Results**). The *comparison* force was delivered at a random interval between 800 ms and 1200 ms after the *test* force. Therefore, in the *no-contact* condition, there was no touch on the right index finger simultaneously with the *test* force on the left index finger.

Before the experiment, participants were trained to make similar movements with their right index finger in the *contact* and *no-contact* conditions, emphasis was placed on restricting their right index finger movements to between the two visual markers in the *no-contact* condition. The 3D position of the right index finger was recorded using a Polhemus Liberty tracker (240Hz). Kinematic information was used to compare the movements between the *contact* and *no-contact* conditions and to reject any trials in which the participant did not trigger the test stimulus with their movement, or trials in which the participant did not move as instructed in the training (see below). Participants were administered white noise through headphones to mask any sounds made by the motor to serve as a cue for the task. The loudness of the white noise was adjusted such that participants could hear the auditory tones of the trial. In all conditions the participants' left index finger as well as the motor that delivered the force to their left index finger, was occluded from vision. This was done to prevent participants using any visual cues to discriminate between the two forces.

Each block consisted of 70 trials resulting in 210 trials per participant. Thus, the proportion of *contact* and *no-contact* trials was the same (50–50%). The order of the conditions was fully counterbalanced across participants.

## Experiment 2

The 2-AFC task was identical to that of Experiment 1. In Experiment 2, there were five conditions in total: the *baseline* condition, the *contact* condition, the *no-contact* condition, and two additional NOGO conditions (*NOGO contact* and *NOGO no-contact*). Trials of the NOGO conditions were pseudo-randomly intermixed with trials of the *contact* and *no-contact* conditions, respectively (GO trials). The five conditions were presented in three separate blocks: the *baseline*, *contact* (GO and NOGO trials), and *no-contact* (GO and NOGO trials) blocks.

In the *contact* block, 50% of trials began with an auditory 'GO' cue (100 ms duration high tone of 2458Hz) instructing participants to tap the active force sensor to trigger the *test* force (identically to the *contact* condition of Experiment 1). On the remaining 50% of trials an auditory 'NOGO' cue (100 ms duration low tone of 222Hz) instructed participants to withhold their movement and the *test* force was then delivered 800 ms following the NOGO cue. The *no-contact* block was identical to the *contact* block, except that the *test* force was triggered by the position of the right index finger without contact with the force sensor (identically to the *no-contact* condition of Experiment 1). The cue tone in the *baseline* block was the same as the cue tone in the NOGO trials (100 ms duration low tone of 222Hz).

Therefore, this design replicated the experimental design of *Thomas et al., 2022* with the additional inclusion of the *baseline* (i.e. externally generated touch) condition. As in Experiment 1, we recorded the 3D position of the right index finger, and the registered kinematic information was used to compare the movements between the *contact* and *no-contact* conditions and reject any trials in which the participants did not trigger the *test* stimulus with their movement, trials in which the participants did not move as instructed in the training (see below), and trials in which the participants moved while instructed not to do so by the auditory cues (NOGO trials). As in Experiment 1, participants were administered white noise and the loudness of the white noise was adjusted such that participants could clearly hear the GO and NOGO auditory cues of the trial.

The *baseline* block consisted of 70 trials, and the *contact* and *no-contact* blocks consisted of 70 GO trials and 70 NOGO trials each, resulting in 350 trials in total per participant. The order of the blocks was fully counterbalanced across participants.

## Experiment 3

The 2-AFC task and the experimental conditions (*baseline*, *contact*, and *no-contact*) were identical to those of Experiment 1, except for the following. *Contact* trials (80%) were now pseudo-randomly intermixed with *no-contact* trials (20%) within the same block. The force sensor was now attached to a plastic platform that could be automatically retracted by a servo motor, depending on the trial type. Upon retraction, a distance sensor placed underneath the platform was revealed. In the *contact* trials, participants tapped the force sensor to trigger the *test* force identically to Experiments 1 and 2. In the *no-contact* trials, the platform was automatically retracted before trial onset, unbeknownst to the participants. This led participants to unexpectedly miss the active force sensor and instead trigger the *test* force only by the position of their right index finger. In all conditions, the participants'

vision was occluded with a blindfold, and white noise was administered via headphones to prevent any visual or auditory cues indicating that the force sensor had been retracted in *no-contact* trials. Note that the blindfold was used for all conditions of Experiment 3 to control for any effects of lack of vision between conditions and allow comparisons between conditions (e.g. *baseline* minus *no-contact*, *no-contact* minus *contact*) with those of Experiments 1 and 2. The *baseline* block was identical to that of Experiments 1 and 2, except for the use of a blindfold.

In Experiment 3, the number of trials was reduced from 70 to 56 to shorten the total experiment time to less than 90 min, similar to Experiment 2. Thus, there were 56 *no-contact* trials (20%) and 224 contact trials (80%). Similarly, there were 56 trials in the *baseline* condition. This resulted in 336 trials per participant. The order of the two blocks was fully counterbalanced across participants.

## Preprocessing of psychophysical trials

### Experiment 1
Experiment 1 included 6300 trials in total (30 participants * 70 trials * 3 conditions). Twenty-nine (29) trials were excluded (0.5%) because of 1 missing response, four trials in which the force was not applied correctly (1.85 N<*test* force <2.15 N), and 24 trials because the *test* force was not triggered when moving towards the distance sensor. The kinematic analysis also excluded all trials excluded by the psychophysical fits.

### Experiment 2
Experiment 2 included 10,500 trials in total (30 participants * 70 trials * 5 conditions). One-hundred fourteen (114) trials were excluded (1.1%) because of two missing responses, six trials in which the force was not applied correctly (1.85 N<*test* force <2.15 N), 44 trials because the *test* force was not triggered when moving towards the distance sensor and 62 trials because the finger moved on a NOGO trial. All trials excluded by the psychophysical fits were also excluded in the kinematic analysis.

### Experiment 3
Experiment 3 included 10,080 trials in total (30 participants * (224 contact trials + 56 no-contact trials + 56 baseline trials)). Two-hundred twenty-four (29) trials were excluded (2.42%) because of 29 missing responses, 86 trials in which the force was not applied correctly (1.85 N<*test* force <2.15 N), 81 because the test force was not triggered when moving towards the distance sensor and 48 because the finger contacted the distance sensor on *no-contact* trials. The kinematic analysis also excluded all trials excluded by the psychophysical fits.

## Preprocessing of kinematic recordings
All kinematic trials were co-registered with the force trials through Transistor-Transistor Logic (TTL) signals sent by the motor to both file outputs. The kinematic recordings were corrected for any distortion in the Polhemus sensor from the force sensor and distance sensor based on measurements made with and without the force/distance sensors within the same space. Exclusion of trials based on the kinematics was done by assessing whether the test force was delivered after the position of the active finger reached its minimum value on the vertical plane. Trials in which the force was delivered after the minimum value had been reached were rejected. For experiment 2, trials in which the active finger moved more than 1 cm following a NOGO cue were also rejected.

## Fitting of psychophysical responses
For each experiment and each condition, the participant's responses were fitted with a generalized linear model using a *logit* link function (**Equation 1**):

$$p = \frac{e^{\beta 0 + \beta 1 x}}{1 + e^{\beta 0 + \beta 1 x}} \tag{1}$$

We extracted two parameters of interest: the Point of Subjective Equality (PSE) ($PSE = \frac{-\beta 0}{\beta 1}$), which represents the intensity at which the *test* force felt as strong as the *comparison* force (p=0.5) and quantifies the perceived intensity, and the Just Noticeable Difference (JND) ($JND = \frac{log(3)}{\beta 1}$), which

reflects the participants' discrimination ability. Before fitting the responses, the values of the applied comparison *forces* were binned to the closest value with respect to their theoretical values (1, 1.5, 1.75, 2, 2.25, 2.5, or 3 N).

In all three experiments, for all participants and all conditions, the fitted logistic models were very good, with McFadden's R-squared measures ranging between 0.602 and 0.979 (*Figure 1—figure supplement 1*, *Figure 2—figure supplement 1*, *Figure 3—figure supplement 1*).

### Normality of data and statistical comparisons

We used R (*R Development Core Team, 2021*), JASP (*Team, 2022*) and MATLAB (2020b) to analyze our data. The normality of the PSE and the JND data distributions, as well as the kinematic information data distributions were assessed using Shapiro-Wilk tests. Depending on the data normality, pairwise comparisons between conditions were performed by using either a paired t-test or a Wilcoxon signed-rank test. We report 95% confidence intervals ($CI^{95}$) for each statistical test. Effect sizes are reported as the Cohen's *d* for t-tests or the matched rank biserial correlation *rrb* for the Wilcoxon signed-rank tests. In addition, a Bayesian factor analysis using default Cauchy priors with a scale of 0.707 was performed for all statistical tests that led to not statistically significant effects, to provide information about the level of support for the null hypothesis compared to the alternative hypothesis ($BF_{01}$) based on the data. We interpret a Bayes factor from 1 to 3 as providing 'anecdotal' support for the null hypothesis, and a Bayes factor from 3 to 10 as providing 'moderate' support for the null hypothesis (*Quintana and Williams, 2018*; *van Doorn et al., 2021*). For the Analysis of Variance (ANOVA), homogeneity of variance was assessed using Levene's test for Equality of Variances, which did not reach significance, and the Q-Q plot of the standardized residuals indicated approximately normally distributed residuals. Post-hoc tests were made using Bonferroni corrections. All tests were two-tailed.

## Acknowledgements

We thank Evridiki Asimakidou and Lili Timar for their assistance during data collection. We also thank Henrik Ehrsson for his helpful comments on an earlier version of the manuscript. KK was supported by the Swedish Research Council (VR Starting Grant 2019–01909). XJ was supported by the Swedish Research Council (VR Starting Grant 2019–01909) and the European Union's Horizon Europe research and innovation programme (Marie Skłodowska-Curie 101059348). Experimental costs were covered by the Swedish Research Council (VR Starting Grant 2019–01909).

## Additional information

### Funding

| Funder | Grant reference number | Author |
| --- | --- | --- |
| Swedish Research Council | 2019–01909 | Xavier Job<br>Konstantina Kilteni |
| HORIZON EUROPE Marie Sklodowska-Curie Actions | 101059348 | Xavier Job |

The funders had no role in study design, data collection and interpretation, or the decision to submit the work for publication.

### Author contributions

Xavier Job, Conceptualization, Formal analysis, Writing – original draft, Writing – review and editing; Konstantina Kilteni, Conceptualization, Writing – original draft, Writing – review and editing

### Author ORCIDs

Xavier Job (ID) http://orcid.org/0000-0002-6017-5983
Konstantina Kilteni (ID) https://orcid.org/0000-0002-6887-6434

## Ethics
Human subjects: All experiments were approved by the Swedish Ethical Review Authority (registration no. 2021-03790). Participants provided written informed consent.

## Decision letter and Author response
Decision letter https://doi.org/10.7554/eLife.90912.sa1
Author response https://doi.org/10.7554/eLife.90912.sa2

---

## Additional files

### Supplementary files
• MDAR checklist

### Data availability
Data (PSEs and JNDs) are publicly available along with pre-registered analyses plans on the OSF (Experiment 1: https://osf.io/9jkqt, Experiment 2: https://osf.io/hs8au, Experiment 3: https://osf.io/pxy87).

The following datasets were generated:

| Author(s) | Year | Dataset title | Dataset URL | Database and Identifier |
|---|---|---|---|---|
| Job X, Kilteni K | 2021 | Attenuation versus enhancement of self-generated touch | https://osf.io/9jkqt | Open Science Framework, 9jkqt |
| Job X, Kilteni K | 2021 | Attenuation of self-generated touch and inhibition | https://osf.io/hs8au | Open Science Framework, hs8au |
| Job X, Kilteni K | 2022 | Attenuation of self-generated touch and active-hand contact | https://osf.io/pxy87/ | Open Science Framework, pxy87 |

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
