## [Editor Report]

While decades of research findings have supported the idea that action attenuates predicted touch, a recent finding has countered this, proposing that action actually enhances predicted touch and that the previously observed attenuation is due to tactile contact. The present study rigorously probes these alternative hypotheses with three pre-registered experiments. They show that previous enhancement findings are due to the lack of a proper baseline condition of externally generated sensation with which to compare. In light of the recent opposing findings, the current paper is important and presents compelling evidence in favor of the sensory attenuation theory.

---

## [Decision Letter]

**Decision letter after peer review:**

Thank you for submitting your article "Action does not enhance but attenuates predicted touch" for consideration by *eLife*. Your article has been reviewed by 3 peer reviewers, one of whom is a member of our Board of Reviewing Editors, and the evaluation has been overseen by Tamar Makin as the Senior Editor.

Essential revisions:

1. Please clarify relative vs absolute claims (as indicated by Reviewer 2).

2. Address Thomas et al.'s Exp 2, where the results suggest additional factors influencing inhibition (as indicated by Reviewer 2).

3. Experiment 3 includes multiple features, such as lack of vision and surprise/distraction, that could also potentially influence perception and lead to attenuation. Please address these potential confounds, and how they may alter the current interpretation of the results.

4. Please clarify that the comparison force was always delivered without involvement of the right hand.

*Reviewer #1 (Recommendations for the authors):*

The experimental manipulation in Experiment 3 for no touch with unexpected no-contact does not only alter predictability. I imagine that there is a large error as the finger accelerated through where the sensor was expected to be. Therefore, it seems possible that the observed attenuation could be due to distraction by the error and unexpected movement. I don't disagree that prediction could play a role, but I would be cautious with that interpretation.

Please explain how the subjects' vision was occluded in Experiment 3. At one point the text says they were blindfolded. Please clarify in the methods.

It would be helpful to show force traces for contact (active and test), no-contact (test) and baseline trials and include as an additional supplemental figure.

It would also be helpful to show movement kinematics for contact and no-contact trials in Experiment 1 and for contact and no-contact trials in Exp 3 since there was no training in experiment 3 (include as an additional supplemental figure).

*Reviewer #2 (Recommendations for the authors):*

The authors should clarify relative vs absolute claims and address Thomas et al.'s Exp 2.

*Reviewer #3 (Recommendations for the authors):*

How have proponents of the enhancement account explained the attenuation evidence? The authors mention that attenuation evidence has been attributed to "unspecific gating processes caused by the simultaneous double tactile stimulation of the two hands" but this is not particularly informative. What did these proponents consider to be special about double tactile stimulation that it should result in the opposite effect on perception as single tactile stimulation? Of course, this is tangential to the present manuscript, but if there is more of a story to this, it may help the reader to know.

Line 143: I didn't immediately realize that the comparison force was always delivered without the involvement of the right hand. It might help others unfamiliar with this paradigm to spell that out.

---

## [Author Response]

Essential revisions:1. Please clarify relative vs absolute claims (as indicated by Reviewer 2).

We have made changes to the manuscript to clarify that our claims about attenuation or enhancement are relative to the baseline condition (Line 207, Lines 287-288, 290-291, 460-461). The importance of the relative claims is also emphasized in Lines 112-125 of our manuscript.

2. Address Thomas et al.'s Exp 2, where the results suggest additional factors influencing inhibition (as indicated by Reviewer 2).

We have provided a detailed response to the point raised by Reviewer 2 regarding the enhancement effect for expected touch when contrasting two conditions, including action (Experiment 2 in Thomas et al., (2022)). In short, based on our findings and those of Bays et al. (2006), we first argue that the touch delivered in Experiment 2 more closely resembled externally generated touch than self-generated touch*.* In addition to the arbitrary mapping between movement and touch (*no-contact* trials), Thomas et al. (2022) also varied the gain between the movement of the active hand and the touch applied on the passive hand on every trial, which made it difficult to establish a causal relationship between movement and touch. Second, we would like to emphasize that in somatosensory research, there is no consensus about how expectations affect perception, even outside the domain of action. For example, Kusnir et al., (2019) recently showed that unexpected touch was perceptually facilitated, in contrast to the findings of Thomas et al., (2022) that expected touch is facilitated. Third, without any baseline condition included, it is hard to interpret the pattern of effects reported in Thomas et al., (2022); for example, are both conditions attenuated but one condition to a lesser extent or is one condition attenuated and one condition perceived accurately? Therefore, we argue that future studies are needed to replicate previous results regarding how expectations affect the perception of externally generated touch.

3. Experiment 3 includes multiple features, such as lack of vision and surprise/distraction, that could also potentially influence perception and lead to attenuation. Please address these potential confounds, and how they may alter the current interpretation of the results.

Regarding the lack of vision, we blindfolded the participants in Experiment 3 to prevent them from seeing the force sensor retract on the (unexpected) *no-contact* trials, which was necessary for building and maintaining the expectation that they would likely contact the force sensor. Importantly, blindfolding participants in Experiment 3 cannot explain any differences between conditions because participants were blindfolded in all conditions of Experiment 3 (*contact*, *no-contact* and *baseline*). Since in the analyses of Experiment 3 (Lines 355-376), we always compared between conditions, any putative effects from blindfolding were effectively removed. Importantly, this also applies to the comparisons that we made between Experiment 3 and Experiments 1 and 2, since all these analyses (Lines 362-376) compared the difference between two conditions (e.g., PSE values for *contact* versus *no-contact* trials or *baseline* versus *no-contact* trials) between the experiments. Once again, any putative effects from blindfolding were effectively removed. Thus, the blindfolding used in Experiment 3 did not alter the interpretation of any of our results.

Regarding the prediction error/surprise/distraction present in the (unexpected) *no-contact* trials of Experiment 3, this was a necessary byproduct of building and violating the expectation that contact would be made on *no-contact* trials. However, it is unlikely that prediction error or surprise *per se* produced attenuation in the *no-contact* condition of Experiment 3 given the literature. First, in our previous study (Kilteni et al., 2019) that used the same force discrimination task and presented touch with or without prediction error using similar proportions of trials as in the present study, we found that participants experienced the more surprising touch to be stronger and more ticklish in the presence of this prediction error, rather than attenuated. Second, previous studies on the effects of surprise/novelty on cognition have shown transient enhancements of perception (Schomaker and Meeter, 2012), facilitated encoding of information into working memory (Mayer et al., 2011) and changes in the allocation of attentional resources in a postdictive manner (for a review of the effects of novelty on cognition, see Schomaker and Meeter, 2015). Interestingly, in the tactile domain, Kusnir et al. (2019) found that tactile stimuli delivered at an unexpected time were perceived with higher sensitivity than the same stimuli delivered at an expected time. Based on this literature, we propose that prediction errors/surprise may act to increase the perceived magnitude, in line with the decreased attenuation that we observed in the *no-contact* condition relative to the *contact* condition of Experiment 3.

We have made changes to the manuscript to refer to these points (Lines 821-827 and 546-554).

4. Please clarify that the comparison force was always delivered without involvement of the right hand.

We have clarified that the comparison force was always delivered without involvement of the right hand (Line 161).

Reviewer #1 (Recommendations for the authors):The experimental manipulation in Experiment 3 for no touch with unexpected no-contact does not only alter predictability. I imagine that there is a large error as the finger accelerated through where the sensor was expected to be. Therefore, it seems possible that the observed attenuation could be due to distraction by the error and unexpected movement. I don't disagree that prediction could play a role, but I would be cautious with that interpretation.

We thank the reviewer for highlighting this. We agree that the unexpected *no-contact* trials in Experiment 3 likely induced some form of prediction error related to the omission of touch on the active hand. For this reason, we raised the possibility in the Discussion that the unexpected omission of touch on the active hand (in *no-contact* trials) could have *postdictive* effects on perception, such as reducing the attenuation of the predicted touch on the passive hand (Lines 526-560). We reasoned that the presence of prediction error is a plausible explanation for why less attenuation was observed for the *no-contact* trials than *contact* trials in that experiment. For example, in one of our previous studies (Kilteni et al., 2019) that used the same force discrimination task and presented touch with or without prediction error using a similar proportion of trials as in the present study, we found that participants experienced the touch to be stronger and more ticklish in the presence of a prediction error, rather than attenuated. Similarly, in the present study, we observed less attenuation in the *no-contact* condition, which can be interpreted as the result of this distraction/prediction error effect. Similarly, research outside the domain of action (Kusnir et al. 2019) showed that perception is improved for tactile events delivered at unpredictable compared to predictable temporal onsets, also suggesting that tactile perception is facilitated in the presence of prediction error.

We have added text to the Discussion to refer to these previous results and explain why prediction error is the likely cause of reduced attenuation in the *no-contact* trials of Experiment 3 (Lines 546-554).

Please explain how the subjects' vision was occluded in Experiment 3. At one point the text says they were blindfolded. Please clarify in the methods.

The participants were blindfolded in Experiment 3. This information has been added to the Methods (Lines 821-827).

It would be helpful to show force traces for contact (active and test), no-contact (test) and baseline trials and include as an additional supplemental figure.

We have included this figure as Figure 3 —figure supplement 3, and we refer to it in the Methods (Lines 727-730).

It would also be helpful to show movement kinematics for contact and no-contact trials in Experiment 1 and for contact and no-contact trials in Exp 3 since there was no training in experiment 3 (include as an additional supplemental figure).

We have included an additional supplemental figure showing the right index finger position data for *contact* and *no-contact* conditions for all three experiments (Figure 3 —figure supplement 2) and refer to it in the Results section (Line 433).

Reviewer #2 (Recommendations for the authors):The authors should clarify relative vs absolute claims and address Thomas et al.'s Exp 2.

Regarding the clarification of relative vs. absolute claims, we have made several changes to the wording of the manuscript to reflect the relative rather than absolute nature of the findings (Line 207, Lines 287-288, 290-291, 460-461). As we wrote in Lines 112-125, we consider the relativity of these claims essential, as the same patterns of results can be incorrectly attributed to “attenuation” or “enhancement” without referring to the reference condition.

Regarding Thomas et al.’s (2022) findings in Experiment 2, we agree that without any (physical, simulated, or expected) contact between the fingers, attenuation is not expected to occur under the classic prediction-based attenuation model since the sensorimotor context necessary for experiencing the touch as self-generated is not present. Our experiments and those of Bays et al. (2006) strongly support this conclusion. We agree with the reviewer that it is interesting that Thomas et al. (2022) found a stronger perceived intensity of touch when it was delivered to the expected finger (i.e., validly cued) compared to the unexpected finger (i.e., invalidly cued). However, we believe that it is difficult to draw conclusions from Thomas et al.’s (2022) findings in Experiment 2 (and Experiment 3) for two reasons.

First, based on our findings, the stimuli delivered by Thomas et al. (2022) more closely resembled exafferent touch (externally generated) that is co-presented with another event (in this case, an action) than *reafferent* touch (self-generated touch)*.* Here, it should be emphasized that in addition to the arbitrary mapping between the movement and the touch (*no-contact* trials), Thomas et al. (2022) additionally varied the intensity of the *test* force (the force that accompanied the movement) on every trial while keeping the comparison force fixed. In other words, every time the participant made the same action (i.e., moved their right index/ring finger), they received a force on their left finger that varied randomly in its magnitude. In contrast, previous studies using force discrimination tasks (Asimakidou et al., 2022; Bays et al., 2006; Kilteni et al., 2021, 2020, 2019; Kilteni and Ehrsson, 2022; Timar et al., 2023) always kept the *test* force constant to establish a perceived causal relationship between the participant’s right index finger movement and touch on the left index finger; that is, every time participants moved their right index finger to press the sensor, they always received the same force on their left index finger. Although we were able to replicate Thomas et al.’s (2022) Experiment 1 findings while also keeping the *test* force fixed in our Experiment 2, it is important to keep in mind that having a variable gain on every trial should decrease attenuation and compromise the experience of the touch as self-generated. Moreover, since the physical magnitude of the tactile stimuli was not reported by Thomas et al. (2022) (i.e., we do not know the numbers of Newtons of the applied forces because they were not measured), it is difficult to assess the impact of a constantly changing gain from trial to trial. Additionally, without any assessment of the perceived intensity of the touch in the absence of action (i.e., a *baseline* condition), it is not possible to interpret the pattern of effects reported in Thomas et al. (2022): for example, are both conditions attenuated but one condition to a lesser extent, is one condition attenuated and one condition perceived accurately, or are both enhanced?

Second, it is also important to note that there is no consensus about how expectations affect somatosensory perception outside the domain of action. Notably, whereas Thomas et al. (2022) showed that expected touch is perceptually enhanced relative to unexpected touch, others have shown that unexpected touch is perceptually enhanced relative to expected touch (Kusnir et al., 2019). Thus, we argue that future studies are needed to test and replicate previous effects of expectations on the perception of externally generated touch.

Reviewer #3 (Recommendations for the authors):How have proponents of the enhancement account explained the attenuation evidence? The authors mention that attenuation evidence has been attributed to "unspecific gating processes caused by the simultaneous double tactile stimulation of the two hands" but this is not particularly informative. What did these proponents consider to be special about double tactile stimulation that it should result in the opposite effect on perception as single tactile stimulation? Of course, this is tangential to the present manuscript, but if there is more of a story to this, it may help the reader to know.

We thank the reviewer for pointing this out. Sensory gating or movement-related gating is a well-documented phenomenon in which *all* tactile inputs applied on a moving limb are suppressed (e.g., have higher detection thresholds) compared to when the limb is at rest (Chapman et al., 1996; Williams et al., 1998; Williams and Chapman, 2000). This mechanism is understood to be distinct from the predictive attenuation of reafferent (i.e., self-generated) tactile input, which operates to attenuate only the input predicted by the movement (for an extensive discussion on the differences between attenuation and gating, please see Kilteni and Ehrsson (2022)). Thomas et al. (2022) attributed previous attenuation effects to ‘unspecific’ or ‘generalized’ gating processes that transfer from the moving hand to the passive hand when the two hands make contact rather than to action prediction. However, to our knowledge, there is no neurophysiological evidence supporting the proposal that generalized gating from the movement of the active hand transfers to passive limbs. In contrast, there is previous experimental evidence demonstrating that gating effects do not transfer to a passive limb (Chapman et al., 1987; Cohen and Starr, 1987; Colino et al., 2014; Papakostopoulos et al., 1975; Pertovaara et al., 1992; Rushton et al., 1981), as discussed on Lines 514-519. Moreover, if attenuation was due to a putative transfer of gating from the active to the passive limb, we would not have observed attenuation in the *no-contact* trials of Experiment 3, where the hands did not make contact.

In our manuscript, we have clarified this point in Line 90 and extended our discussion of this point in Lines 578-580.

Line 143: I didn't immediately realize that the comparison force was always delivered without the involvement of the right hand. It might help others unfamiliar with this paradigm to spell that out.

We thank the reviewer for pointing this out. We have made this explicit (Line 161).

References:

Asimakidou E, Job X, Kilteni K. 2022. The positive dimension of schizotypy is associated with a reduced attenuation and precision of self-generated touch. Schizophrenia 8. doi:10.1038/s41537-022-00264-6

Bays PM, Flanagan JR, Wolpert DM. 2006. Attenuation of self-generated tactile sensations is predictive, not postdictive. PLoS Biol 4. doi:10.1371/journal.pbio.0040028

Bays PM, Wolpert DM. 2008. Predictive attenuation in the perception of touch. Sensorimotor Foundations of Higher Cognition. doi:10.1093/acprof:oso/9780199231447.003.0016

Chapman CE, Bushnell MC, Miron D, Duncan GH, Lund JP. 1987. Sensory perception during movement in man. Exp Brain Res 68:516–524. doi:10.1007/BF00249795

Chapman CE, Zompa IC, Williams SR, Shenasa J, Jiang W. 1996. Factors influencing the perception of tactile stimuli during movement. Somesthesis and the Neurobiology of the Somatosensory Cortex. doi:10.1007/978-3-0348-9016-8_25

Cohen LG, Starr A. 1987. Localization, timing and specificity of gating of somatosensory evoked potentials during active movement in man. Brain. doi:10.1093/brain/110.2.451

Colino FL, Buckingham G, Cheng DT, van Donkelaar P, Binsted G. 2014. Tactile gating in a reaching and grasping task. Physiol Rep 2:1–11. doi:10.1002/phy2.267

Kilteni K. 2023. Methods of Somatosensory Attenuation. Somatosensory Research Methods. New York : Springer US. pp. 35–53. doi:10.1007/978-1-0716-3068-6_2

Kilteni K, Ehrsson HH. 2022. Predictive attenuation of touch and tactile gating are distinct perceptual phenomena. iScience 25. doi:10.1016/j.isci.2022.104077

Kilteni K, Engeler P, Boberg I, Maurex L, Ehrsson HH. 2021. No evidence for somatosensory attenuation during action observation of self-touch. Eur J Neurosci 54:6422–6444. doi:10.1111/ejn.15436

Kilteni K, Engeler P, Ehrsson HH. 2020. Efference Copy Is Necessary for the Attenuation of Self-Generated Touch. iScience 23. doi:10.1016/j.isci.2020.100843

Kilteni K, Houborg C, Ehrsson HH. 2019. Rapid learning and unlearning of predicted sensory delays in self-generated touch. eLife 8. doi:10.7554/eLife.42888

Kusnir F, Pesin S, Moscona G, Landau AN. 2019. When temporal certainty doesn’t help. J Cogn Neurosci 32. doi:10.1162/jocn_a_01482

Mayer JS, Kim J, Park S. 2011. Enhancing visual working memory encoding: The role of target novelty. Vis cogn 19. doi:10.1080/13506285.2011.594459

Papakostopoulos D, Cooper R, Crow HJ. 1975. Inhibition of cortical evoked potentials and sensation by self-initiated movement in man. Nature 258:321–324. doi:10.1038/258321a0

Pertovaara A, Kemppainen P, Leppänen H. 1992. Lowered cutaneous sensitivity to nonpainful electrical stimulation during isometric exercise in humans. Exp Brain Res 89:447–452. doi:10.1007/BF00228261

Rushton DN, Roghwell JC, Craggs MD. 1981. Gating of somatosensory evoked potentials during different kinds of movement in man. Brain 104:465–491. doi:10.1093/brain/104.3.465

Schomaker J, Meeter M. 2015. Short- and long-lasting consequences of novelty, deviance and surprise on brain and cognition. Neurosci Biobehav Rev. doi:10.1016/j.neubiorev.2015.05.002

Schomaker J, Meeter M. 2012. Novelty enhances visual perception. PLoS One 7:e50599.

Shergill SS, Bays PM, Frith CD, Wolpert DM. 2003. Two eyes for an eye: the neuroscience of force escalation. Science 301:187. doi:10.1126/science.1085327

Shergill SS, Samson G, Bays PM, Frith CD, Wolpert DM. 2005. Evidence for sensory prediction deficits in schizophrenia. Am J Psychiatry 162:2384–2386. doi:10.1176/appi.ajp.162.12.2384

Teufel C, Kingdon A, Ingram JN, Wolpert DM, Fletcher PC. 2010. Deficits in sensory prediction are related to delusional ideation in healthy individuals. Neuropsychologia 48. doi:10.1016/j.neuropsychologia.2010.10.024

Thomas ER, Yon D, de Lange FP, Press C. 2022. Action Enhances Predicted Touch. Psychol Sci 33. doi:10.1177/09567976211017505

Timar L, Job X, Orban de Xivry J-J, Kilteni K. 2023. Aging exerts a limited influence on the perception of self-generated and externally generated touch. J Neurophysiol. doi:10.1152/jn.00145.2023

Williams SR, Chapman CE. 2000. Time course and magnitude of movement-related gating of tactile detection in humans. II. Effects of stimulus intensity. J Neurophysiol 84. doi:10.1152/jn.2000.84.2.863

Williams SR, Shenasa J, Chapman CE. 1998. Time course and magnitude of movement-related gating of tactile detection in humans. I. Importance of stimulus location. J Neurophysiol 79:947–963. doi:10.1152/jn.00527.2001

Wolpe N, Ingram JN, Tsvetanov KA, Geerligs L, Kievit RA, Henson RN, Wolpert DM, Rowe JB. 2016. Ageing increases reliance on sensorimotor prediction through structural and functional differences in frontostriatal circuits. Nat Commun 7:13034. doi:10.1038/ncomms13034